# The Promise of Nanoparticles-Based Radiotherapy in Cancer Treatment

**DOI:** 10.3390/cancers15061892

**Published:** 2023-03-22

**Authors:** Munima Haque, Md Salman Shakil, Kazi Mustafa Mahmud

**Affiliations:** 1Department of Mathematics and Natural Sciences, BRAC University, Dhaka 1212, Bangladesh; 2Department of Biochemistry and Molecular Biology, Jahangirnagar University, Savar, Dhaka 1342, Bangladesh

**Keywords:** cancer treatment, radiotherapy, radiation treatment, nanoparticles, nanoparticle-based radiotherapy, combination therapy, phototherapy, mechanism

## Abstract

**Simple Summary:**

Radiotherapy (RT) is used worldwide as a gold standard treatment approach for cancer management. However, the RT treatment modality contains limitations along with numerous side effects. Nanoparticles (NPs) have unique properties that can be utilized in the field of cancer treatment. Therefore, the combination of NPs with RT opens a new arena in cancer treatment. Their synergistic effect strengthens ionizing radiation sensitivity and allows for tumor-selective treatment while reducing side effects. More importantly, NP-based RT offers greater control over RT alone and has shown higher selectivity. In addition, the combined treatment also helps to overcome radioresistance and drug-resistance phenomena. The main mechanism through which NP-based RT destroys cancer cells includes production of ROS, which damage DNA, inhibiting the DNA-repair system, perturbing the cell cycle, and controlling the tumor microenvironment. NP-based RT has been reported to destroy cancer stem cells and has shown good results in clinical trials. Moreover, the addition of phototherapy to NP-based RT reduces the limitations of phototherapy and has shown excellent cancer cell-killing potentiality.

**Abstract:**

Radiation has been utilized for a long time for the treatment of cancer patients. However, radiotherapy (RT) has many constraints, among which non-selectivity is the primary one. The implementation of nanoparticles (NPs) with RT not only localizes radiation in targeted tissue but also provides significant tumoricidal effect(s) compared to radiation alone. NPs can be functionalized with both biomolecules and therapeutic agents, and their combination significantly reduces the side effects of RT. NP-based RT destroys cancer cells through multiple mechanisms, including ROS generation, which in turn damages DNA and other cellular organelles, inhibiting of the DNA double-strand damage-repair system, obstructing of the cell cycle, regulating of the tumor microenvironment, and killing of cancer stem cells. Furthermore, such combined treatments overcome radioresistance and drug resistance to chemotherapy. Additionally, NP-based RT in combined treatments have shown synergistic therapeutic benefit(s) and enhanced the therapeutic window. Furthermore, a combination of phototherapy, i.e., photodynamic therapy and photothermal therapy with NP-based RT, not only reduces phototoxicity but also offers excellent therapeutic benefits. Moreover, using NPs with RT has shown promise in cancer treatment and shown excellent therapeutic outcomes in clinical trials. Therefore, extensive research in this field will pave the way toward improved RT in cancer treatment.

## 1. Introduction

Cancer is a great threat to global public health [1]. It is one of the main causes of significant morbidity and global deaths per year [2]. In 2022, approximately 1,918,030 cancer cases and 609,360 cancer related deaths were reported [3], and 26 million new cancer cases are projected to occur by 2030 [4]. Cancer develops by uncontrolled proliferation of cells due to pathophysiological alterations of the inherent cell division process that later disseminate to different tissues [5,6]. Radiotherapy (RT) is actively used to treat primary, as well as terminal, tumors [7]. This course of treatment damages cancer cells upon exposure to ionization radiation [8]. A beam of high-energy ionization radiation destroys intracellular components of tumors, killing cancer cells [9]. Radiation treatment is effectual for more than half of cancer patients [10,11,12] and is applied in two-thirds of cancer treatment regimens in Western countries as a significant treatment modality for locoregional tumors [13]. Approximately 30–50% of all cancer patients receive RT, either alone or as an adjuvant treatment [14].

However, RT has limitations that includes dose heterogeneity, intrinsic as well as acquired resistance to RT, and tumor recurrence [15,16]. In addition, delivering sufficient tumoricidal radiation doses induces toxicity in both cancer and nearby non-cancerous cells since both cells have similar mass energy absorption tendencies [17]. Therefore, RT is restricted by the highest tolerated dose to the surrounding normal tissues [13]. Moreover, hypoxic tumors resist radiation treatment [18]. RT also exhibits side effects (acute, consequential, or late complications) causing radiation toxicity in the skin, mucosa, liver, lungs, kidneys, and heart [19,20,21,22,23].

The advent of nanoparticles (NPs) has assisted in the evolution of traditional RT from a “one-size-fits-all” concept to tailored and dynamic treatment modalities [13]. High surface-to-volume ratios, increased cellular uptake, and adjustability make NPs a suitable choice for tumor-targeted and less toxic treatment approaches [24,25,26,27,28,29]. NPs that are prepared from high Z materials act as radiosensitizers, receiving external beams of ionizing radiation [30,31]. Additionally, NPs can be modified by target molecules, and radiosensitizers are able to penetrate and accumulate in tumors to achieve tumor selectivity, as well as enhanced therapeutic doses [32,33,34,35]. Such NP-based radiosensitizing agents draw contrasts between cancer and non-cancerous cells owing to variability in their mass absorption coefficients [17]. Nanostructured radiosensitizers intensify radiation to increase the local dose and overcome hypoxia and rapid proliferation [36,37]. The combination of phototherapy with NP-based RT further improvises cancer management by increasing significant anticancer activity while reducing the limitations of both photothermal and photodynamic treatment [38].

In this review, we highlight the potentiality of NP-based RT to overcome the limitations of conventional RT, as well as combined treatment of RT with chemotherapy (CT). We have describe the robust and significant tumoricidal activity of the NP-based radiotherapeutic approaches compared to radiation or NP treatment only in cancer, and we explore its effects on cancer stem cells and in clinical trials. In addition, NP-based RT’s potential to overcome radioresistance and drug resistance is discussed. Moreover, we indicate some crucial limitations of NP-based RT and address the future prospects of this treatment modality.

## 2. Combinational Use of RT and Chemotherapy

Radiotherapy utilizes high-energy beams to destroy cancer cells and thereby shrink tumors [39]. On the other hand, chemotherapy (CT) uses cytotoxic drugs that are capable of killing cancer cells and inhibit cancer cell growth [40]. These conventional treatments have limitations, such as insufficient therapeutic properties and side effects [41]. In this regard, combination of RT with CT offers an improved therapeutic effect compared to using a single approach. CT is widely used in lung cancer [42], esophageal cancer [43], rectal cancer [44], and hepatocellular carcinoma [45]. However, toxic side effects and a lack of selectivity and synergy between RT and CT are key problems in chemotherapeutic treatment. In addition, severe side effects of Pt-based anticancer drugs have restricted their clinical application [41,46]. On the other hand, RT is often off-target and damages surrounding healthy cells, and it is difficult to obtain optimum radiation [47]. The application of nanostructured radiosensitizers in RT has attained great attention recently. Using high Z materials as nano radiosensitizers enhances RT owing to their Compton scattering and photoelectric effects [48]. Moreover, introducing platinum-based anticancer drugs, such as cisplatin (II), oxaliplatin (II), and carboplatin (II), as radiosensitizers has yielded more effective RT strategies [49].

A key advantage to combined treatment is the prospect of achieving a greater organ-preservation rate. Another advantage is its independent cell-destroying effect. RT aims to control the primary tumor, whereas CT eliminates distant metastasis. The combined treatment modality is also advocated based on clinical trial results. Phase II trials have reported convincing results of combined treatments [50]. However, the combined approach has some challenges to overcome. Several parameters, such as the dose, duration, administration sequence, should be optimized properly [51]. Therefore, further research is needed to explore these limitations.

## 3. How Radiation Reacts with Radiosensitizers of High Z Materials

The bombarding of ionizing radiation with NPs gives rise to several outcomes, including photoelectric effects, the Compton electron effect, and Auger electron effects (Figure 1) [40]. The radiation energy is imparted to the electrons of the NPs’ atoms, causing the ejection of electrons from their orbits [41]. Such electronic ejection occurs with a kinetic energy that is equivalent to the radiation wave energy minus the binding energy of the electrons and assesses the electron range in tissue [42].

Photoelectric effects occur when low-energy photons interacts with materials (<60 keV). The photon energy is exclusively absorbed by inner orbital electrons and is ejected from its orbit. This phenomenon causes the electrons of outer orbits to shift to inner orbits and empty space. Thus, liberated fluorescence photons with specific wavelengths depend on the difference between the energy of two orbits, called secondary radiation (Figure 1). Later, Auger electrons are emitted when outer orbit electrons fill the empty space of the inner orbit due to photoelectric effects. This process relinquishes energy to outer orbits electrons, leading to ejection of electrons from higher orbits (Figure 1) [17]. The Auger electrons have high linear energy transfer properties and hence could be extremely injurious to cells [43]. The probability of photoelectric effects occurring is assessed by the formula (Z/E)^3^, where E = the incoming photon’s energy, and Z = the absorber molecule’s atomic number. Thus, the possibility of photoelectric effects is enhanced with increased absorber atoms, but it decreases with increased energy of incident photons. Photoelectric effects contribute more to radiation interaction with high Z metal NPs than to absorption in soft tissue. Therefore, photoelectrons, secondary photons, and Auger electrons released from high-Z metal NPs enhance localized doses, along with focal ionization of nearby cells via photoelectric effects. Since photoelectric effects decrease with increased energy of photons, most of the nanoparticles combining with radiation treatment use keV photons to optimize the radiosensitization and enhance the local dose by 10–150 times [44,45,46].

The Compton interaction dominates within 25 keV–25 MeV of photon energy. Since most RT is performed at an energy level of 6–20 MeV, this effect is the most common interaction between incident photons and cancer tissue. In the case of the Compton effect, incident photons strike weakly bound outer orbit electrons and donate part of their energy to the electrons, stimulating electrons to leave the outer orbit. Concurrently, the photons become scattered after giving part of their energy and further interacting with other atoms (Figure 1). Afterward, the emitted electrons continue to ionize adjacent tissues. The possibility of a Compton interaction depends inversely on the incoming photon’s energy but is not dependent on the material’s atomic number. Therefore, high-Z metal NPs do not have a substantial role in the Compton effect [17,40,46,47].

## 4. Biological Response of NPs-Based RT

Complementing nanotherapeutics with ionizing radiation exhibits an enhanced biological response in cancer management through several approaches. The major mechanisms include inhibiting DNA-repair processes, producing reactive oxygen species (ROS), which damage DNA or other biomolecules by oxidation, inhibiting tumor metastasis by controlling the tumor microenvironment (TME), and arresting the cell cycle (Figure 2).

### 4.1. DNA Damage

Ionizing with X-rays, γ-rays, or proton radiation itself causes spontaneous double strand breaks (DSBs) in DNA by breaking atomic and molecular bonds [48,49,50], and un-repairing of DSB leads to genetic instability, cell division termination, and reduced proliferation and consequently to death [51,52]. However, such DSBs are repaired by the cellular DNA damage response (DDR) [53]. Three main proteins from the phosphatidylinositol 3-kinase-related kinase family, namely ataxia-telangiectasia mutated (ATM), ATM and Rad3 related (ATR), and DNA-dependent protein kinase (DNA-PK), are involved in identifying and repairing DNA DSBs [54]. Three different types of sensor protein complexes are responsible for the recruitment, as well as activation, of these three proteins of the PI3K family at damaged DNA sites, i.e., MRE11/RAD50/NBS1 (MRN) for ATM, RPA/ATRIP for ATR, and KU70–KU80/86 for DNA-PK [55].

The main goal of DDR machinery is to delay the progression of the cell cycle and fix the damage [56]. When cancer cells are exposed to IR, they undergo transient cell cycle arrest to repair DSBs either by non-homologous end joining (NHEJ) throughout the entire cell cycle or by homologous recombination (HR) during the S and G2 phase [57]. ATM is primarily responsible for activating the HR pathway. Other proteins, including breast and ovarian susceptibility protein (Brca2), Rad51, and X-ray repair cross complementing protein 2 (XRCC2), are also involved in the HR pathway [58]. On the other hand, DNA-PK mainly regulates the NHEJ pathway in association with DNA ligase IV and X-ray repair cross complementing protein 4 [59,60]. Another pathway involves both ATM and ATR. Here, MRN sensor protein complex senses damage sites and activates ATM [61]. Autophosphorylation of ATM kinase sends signals to transducers such as checkpoint kinase 2 (Chk2) and the transcription factor p53. p53 controls the expression of p21, which interacts with cyclin-dependent kinase (CDK) complex and arrests the G1 phase of the cell cycle [62]. Modification of chromatin also occurs together with the process, and then the DNA repair process is initiated. However, mutation of the p53 makes the G1 checkpoint defective in most of the cancer cells. Hence, the G2 checkpoint plays a crucial role for surviving cancer cells. RPA/ATRIP sensor protein complex recognizes and activates ATR. Here, ATR phosphorylates checkpoint kinase 1 (Chk1), which degrades cell division cycle 25A (CDC25A) through further phosphorylation and slows the progression of DNA replication during the S phase. The ATM-Chk2 and ATR-Chk1 signaling pathway acts together with DNA-PK phosphorylate p53, which controls genes required for DNA repair, arresting the cell cycle, and apoptosis [55,61,63].

For this purpose, DNA double-strand repair inhibition (DSBRI) appears to a promising strategy for RT [64]. However, it is a challenge to achieve tumor selective DSBRI-based radiotherapeutic treatment since such approaches often sensitize normal cells [65]. Interestingly, introduction of NP-based radiotherapeutic approaches mediates tumor-specific DSBRI owing to their increased permeability and retention effects [66,67]. Zhang et al. developed nano-constructure by combining androgen receptor (AR) with shRNA and folate-targeted H1 nanopolymer (NP AR-shRNA). NP AR-shRNA selectively destroyed prostate cancer cells by mimicking DNA DSBs and activated kinase activity, in turn impeding DNA damage repair signaling pathways (Figure 2) [68]. The presence of γ-H2AX is used as an indicator to detect DNA DSB in higher eukaryotes [69,70]. NP AR-shRNA in combination with IR (4 Gy) increases the expression of γ-H2AX in PC3 and 22Rv1 cells by nearly three-fold compared to IR alone (4 Gy). In vivo experiments showed that irradiation downregulated the expression of AR protein while increasing γ-H2AX expression in 22Rv1 tumors. Additionally, mice bearing PC3 and 22Rv1 cells were exposed to NP AR-shRNA under X-ray (total 9 Gy) and displayed significant tumor reduction compared to only X-ray treatment [68]. Similarly, Yao et al. formulated nanoparticles by conjugating DSB bait (Dbait) with H1 polymer (Dbait@H1 NPs), selectively killing prostate cancer cells upon exposure to radiation by inhibiting DSB repair [71]. Dbait@H1 NPs attached selectively to folate receptor and mimicked DNA DSBs upon release into the nucleus [72,73]. Dbait of Dbait@H1 NPs activates DNA-PK and phosphorylate γ-H2AX. Then, factors associated with DNA damage repair are assembled at the free end of Dbait, preventing them from affecting the DSB sites of the real chromosome and resulting in prolonged defects in DSB repair [74,75,76,77]. Irradiating (4 Gy) PC-3 cells with Dbait@H1 NPs induced γ-H2AX foci numbers three times higher than radiation alone (4 Gy) (Table 1). Moreover, the same radiation dose also increases phosphorylation of DNA-PK and H2AX of both PC-3 and 22Rv1 cells. Moreover, an in vivo study showed that mice carrying both PC-3 and 22Rv1 cells were exposed to 9 Gy of radiation and 60 μg/kg of Dbait@H1 NPs and exhibited 1.67- and 2.5-fold reduced tumor volumes, respectively, compared to only radiation (9 Gy) (Table 1) [71]. Similarly, HeLa cells exposed to AuNPs and irradiation at 4 Gy of 220 kVp and 6 MVp caused enhanced γ-H2AX, suggesting induction of possible DNA DSB [78]. Combined treatment with PEGylated-AuNPs and 4 Gy RT (150 kVp) increased DNA damage 1.7-fold in U251 cells compared to radiation alone [79]. Additionally, Zheng et al. reported that AuNPs at a radiation dose of 6 Gy induced DSB in HepG2 cells [80].

Chen et al. modified AuNPs with bovine serum albumin (BSA@AuNPs), which induced a 2.02-fold increase in γ-H2AX density compared to X-ray radiation only in U87 cells upon exposure to a 3-Gy dose of 160 kVp X-ray. Moreover, treating mice with BSA@AuNPs under X-ray radiation (5 Gy) reduced tumor volume significantly compared to X-ray radiation alone [81]. Similarly, a nanoformulation of KU55933 (NPs@KU55933) impeded the repair process of DSBs and exhibited enhanced tumor volume reduction in vivo. Exposing two types of lung cancer cells (i.e., H460 and A549 cells) to NPs@KU55933 at a radiation dose of 15 Gy showed a remarkable tumor size reduction compared to the X-ray irradiation group (Table 1) [82].

### 4.2. Reactive Oxygen Species (ROS)

Ionization in combination with NPs brings about indirect necrosis or apoptosis via oxidation of biomolecules, including proteins, lipids, and DNA, along with mitochondrial dysfunction [7,83,84]. High-intensity ionizing radiation generates ROS, including superoxide anion radicals (O_2_^−^), hydrogen peroxide (H_2_O_2_), and hydroxyl radicals (^•^OH), through water radiolysis, and they interact with cellular biomolecules, leading to apoptosis/cellular death [7,85,86,87,88,89], as well as suppressing tumor progression [90]. Therefore, sufficient ROS production is crucial to mediate DNA damage, as well as suppress DNA repair (Figure 2).

Zhao et al. fabricated Gd-bearing polyoxometalates linked with chitosan nanospheres and integrated with hypoxia inducible factor 1α (HIF-1α) siRNA (GdW10@CS_HsiRNA) that showed enhanced radiosensitization in hypoxic tumors. GdW10@CS functions as an external radiosensitizer for depositing ionizing radiation doses and as a nanocarrier of HIF-1α siRNA to stop DNA DSB restoration. Additionally, GdW10@CS annihilates intracellular reduced glutathione (GSH) levels upon exposure to X-ray radiation, leading to overproduction of ROS through W6^+^-triggered GSH oxidation, thereby facilitating radiotherapeutic efficiency. Irradiating BEL-7402 cells with 6 Gy of X-ray and GdW10@CS (100 μM) produced Compton and Auger electrons, which interact with surrounding H_2_O or O_2_ molecules, thus producing 10-fold more ROS, as well as exhausting GSH levels three times more compared to X-ray treatment alone. Furthermore, 20 μL of GdW10@CS along with 10 Gy of X-ray radiation were administered into BALB/c mice bearing BEL-7402 tumors and reduced tumor volumes by nearly five- and eight-fold compared to GdW10@CS_HsiRNA without RT and RT without GdW10@CS_HsiRNA, respectively [91]. Zhan et al. designed a nano-enabled coordination platform with bismuth and cisplatin prodrug (NP@PVP) that improves the efficiency of chemoradiotherapy by X-ray radiation. The bismuth in NP@PVP increases generation of ROS to intensify DNA damage after X-ray irradiation. Treating EMT-6 cells with NP@PVP and X-ray irradiation (5 Gy) generated 3.21-fold more ROS compared to platinum-based drugs along with X-ray irradiation at the same dose. In vivo results showed that mice carrying EMT-6 tumors treated with NP@PVP (2 mg kg^−1^) under irradiation with X-rays (5 Gy) showed 54.7% inhibited tumor growth compared to X-ray treatment only [92]. Choi et al. synthesized radiation-responsive PEGylated gold nanoparticles containing dihydrohodamine 123 (DHR-123) (RPAuNPs). RPAuNPs absorbs the X-ray energy and transfers it to nearby molecules through electrons, including photoelectrons, Compton election, and Auger electrons, which generate ROS by water radiolysis. Exposure of RPAuNPs (7.75 μg/mL) at to Gy of X-ray radiation increases the fluorescence by 7-fold owing to the local generation of ROS. The same treatment of MDA-MB-231 cells yields similar results, with increased ROS levels in nearby cells. Mice bearing MDA-MB-231 xenografts treated with RPAuNPs at a concentration of 6 μg/100 μL and irradiation of 6 Gy at 225 kVp exhibited 3- to 6-fold higher ROS production compared to the treatment with RPAuNPs without X-ray irradiation [93]. In addition, mitochondria are an important source of cellular ROS production [94]. Tang et al. constructed Gd-doped titanium dioxide nanosensitizer (G@TiO2 NPs), which targets mitochondria for effective RT. G@TiO2 NPs generates ROS effectively since it possesses a large photoelectric cross-section for X-rays [95].

### 4.3. Tumor Microenvironment (TME)

The tumor microenvironment (TME) consists of various types of cells (endothelial cells, immune cells, fibroblasts, etc.) and extracellular components (extracellular matrix, growth factors, cytokines, etc.) that surrounds tumors and are nourished by blood and the lymphatic vascular network [96,97]. The TME functions critically in the regulation of tumor progression, immune escape, and metastasis [98]. It has significant influence on therapeutic effectivity [99]. Therefore, TME-associated NP-based RT offers potentiality in destroying cancer cells efficiently [100].

ROS are among the key players to alternate the tumor microenvironment (TME) during RT [101]. Among many other features of the TME, hypoxia mostly compromises tumor sensitivity to anticancer drugs, along with reactivity toward free radicals, thus creating hurdles in RT [7]. The anoxic and hypoxic TME of solid tumors compromises the production of ROS, causing poor responses to tumor cells [102,103]. Furthermore, hypoxia=activated hypoxia inducing factor-1 promotes resistance to RT and increases expression of genes involved in angiogenesis and metastasis of tumors [104,105,106]. Therefore, management of the TME is critical in killing cancer cells.

O_2_ is indispensable during RT since it reacts with DNA breaks to avoid repair of DNA by tumor cells, thus relieving hypoxia and enhancing RT-mediated cell killing [107]. Considering this point, Liu et al. prepared a nanostructure system, PFC@PLGA-RBCM, by enveloping perfluorocarbon (PFC) within poly(d,l-lactide-co-glycolide) (PLGA), which then was coated with a red-blood-cell membrane (RBCM). PFC@PLGA-RBCM NPs contains a PFC core, which is able to dissolve O_2_ to a great extent, and the RBCM coating contributes to enhanced blood circulation of NPs. PFC@PLGA-RBCM NPs deliver oxygen efficiently to the TME, which helps to relieve tumor hypoxia and enhance the efficacy of RT. Injecting 4T1-tumor-bearing mice with PFC@PLGA-RBCM NPs (200 µL) and exposing them to X-ray radiation (8 Gy) showed enhanced tumor volume reduction by nearly 8- and 2.5-fold compared to PFC@PLGA-RBCM NPs without irradiation and X-ray irradiation only, respectively [108]. Chen et al. fabricated NPs via encapsulating catalase (Cat) by poly(lactic-co-glycolic) acid and hydrophobic imiquimod (Cat@PLGA_R837). Upon irradiation (8 Gy) in CT-26 cell lines, Cat loaded inside Cat@PLGA_R837 decomposes H_2_O_2_ to produce O_2_, thus relieve the hypoxic TME significantly (Table 1) [89]. Moreover, X-ray irradiation induced gold NPs with silica cores (SAuNPs) to exhibit enhanced antitumor effects under a hypoxic environment. At 8 Gy of radiation, SAuNPs caused 20% more cellular death of CT26 cells than in a control group under hypoxic conditions, while only 5% death occurred under normoxic condition. In addition, irradiation with X-rays increased ROS production by 40% in normoxic conditions compared to 20% in hypoxic conditions [109].

### 4.4. Targeting the Cell Cycle

Targeting the cell cycle is an effective approach in cancer treatment. Different nano formulations in combination with radiation mediate disruption of the cell cycle, leading to apoptosis [85]. Furthermore, cell cycle phases have distinctive effects on radiosensitivity. For instance, the late S-phase is the most radioresistant, while G2 is the most radiosensitive phase [110]. Cells activate cell cycle checkpoints in the G1, S, and G2 phases in response to radiation to repair genomic defects, maintenance integrity, or prevent cell division through activation of cell death mechanisms [111]. The literatures has reported that NPs with radiation arrest mostly the G2/M phase while reducing cells in the G0/G1 phase of the cell cycle (Figure 2) [112,113,114].

Roa et al. constructed gold NPs capped with glucose (Glu-AuNPs), showing improved cell-targeting capacity and excellent radio-sensitization. Glu-AuNPs stimulated activation of cyclin dependent kinase (CDK), leading to an accelerated G0/G1 phase and halting the G2/M phase of the cell cycle by activating CDK1 and CDK2. Irradiating DU-145 cells with 2 Gy of ortho-voltage together with Glu-AuNPs (15 nM) arrested the cell cycle at the G2/M phase and exhibited enhanced growth inhibition by 1.5- to 2-fold compared to X-ray alone. Glu-GNPs inhibited cyclin A’s expression by 42.3%. Cyclin A together with CDK2 form cyclin A–CDK2 complex and initiate the transition of G2/M. Hence, inhibiting cyclin A caused G2/M transition delay [115]. Chen et al. synthesized ultra-small selenium NPs (SeNPs) of 27.5 nm by chemical methods [116]. SeNPs have shown excellent biological activity and low toxicity [117,118]. Upon irradiation, SeNPs arrested the G2/M phase while accelerating the G1/S phase. X-ray irradiation of SeNPs (0.15 μg/mL) at 6 Gy on MCF-7 cells upsurged the G2/M phase proportion by 7.4-fold compared to radiation alone [116]. Xu et al. conjugated gold with glycine (G), arginine (R), and aspartate (D) peptides (Au-G-R-D). While exposed to 6-mV X-rays with a 4-Gy dose, Au-G-R-D (50 μg/mL) arrested the G2/M phase significantly (6.4%) in A375 melanoma cells compared to radiation alone [119].

**Table 1 cancers-15-01892-t001:** In vivo effects of NP-based RT.

Nanoparticle Formulation	Test Animal Model	Animal Age	Cancer Type	Cell Line	Tumor Growth Delay Compared to Control Group				Concentration	Radiation Dose	Observations	Ref.
					RT	Nanoparticle	Nanoparticle + RT	Statistical Significance Method				
KU55933	Nu/Nu mice	6–8 weeks	NSCL	H460	Significant	Significant	Significant	Response features analysis method	500 mg/kg	15 Gy	Impedes the repair process of DNA double-stranded breaks	[82]
				A549	Significant	Significant	Significant					
Dbait@H1 NPs	Male nude mice	4–6 weeks	PCa	PC-3	Significant *p* = 0.032	NR	Significant *p* = 0.004	Two-sample *t*-tests and analysis of variance	60 μg/kg	9 Gy	Signaling pathways for repairing DNA damage are inhibited by Dbait@H1 NPs, which are competitive inhibitors of DSB.	[71]
				22Rv1	Significant *p* = 0.001	NR	Significant *p* < 0.001					
Cat@PLGA_R837	Balb/c mice	NR	CRC	CT-26	No appreciable inhibition of tumor growth	NR	Significantly suppresses tumor growth	One-way ANOVA using the Tukey’s post-test	Cat = 0.5 mg/kg; R837 = 0.6 mg/kg	8 Gy	Decomposes H_2_O_2_ to produce O_2_ in the tumor microenvironment.	[120]
IPI549@HMP	Balb/c mice	6–8 weeks	CRC	Luc+ CT26 cells	Significant	Moderately significant	Significant	Student’s two-tailed unpaired *t*-test	MnO_2_ = 7.5 mg/kg; IPI549 = 1.5 mg/kg	6 Gy	Myeloid cells are selectively targeted by IPI549@HMP and decompose endogenous H_2_O_2_ to O_2_ upon X-ray irradiation	[121]
AuNPs	Balb/C mice	NR	MC	EMT-6	Non-significant	Significantly delayed tumor growth but did not reduce tumor volume significantly	Significantly decreased tumor growth and tumor volume	Wilcoxon’s non-parametric two-sample rank-sum	1.35 g/kg	30 Gy	AuNPs, being high-Z nanoparticles, preferentially absorb X-rays and subsequently reduce tumor.	[122]
NPs-lncAFAP1-AS1 siRNA	BALB/c normal mice and nude mice	4–5 weeks	BC	MDA-MBA-231R	Significantly decreased tumor volume *p* < 0.001	Significantly decreased tumor volume *p* < 0.01	Significantly decreased tumor volume *p* < 0.001	Student’s unpaired two-sided *t*-test and one-way ANOVA	1 nmol siRNA	10 Gy	Tumor growth is reduced due to blockage of the Wnt/β Catenin signaling pathway and scavenging intracellular GSH.	[123]
Au@Tat-R-EK NPs	BALB/c mice	6–8 weeks	LC	LM3	Moderately significant	Not Significant	Significant	One-way analysis of variance (ANOVA) in Origin software	25 mg/kg	6 Gy	Au@Tat-R-EK NPs respond to overexpressed cathepsin B in the tumor microenvironment that induces site-specific enhancement of tumor cell uptake and afterward damages DNA effectively upon X-ray irradiation.	[124]
AuNP@CRZ	Nude mice	10–11 weeks	ACC of the salivary gland	ACC	Impeded tumor growth (*p* < 0.001) but did not significantly reduce tumor volume to less than its size at T0 (*p* > 0.05)	Significantly decreased tumor volume (*p* < 0.001) and caused a reduction in tumor volume (*p* < 0.001).	Decreased tumor volume compared to previously (*p* < 0.001), and it caused significant shrinkage of the tumor to near disappearance (*p* = 0.007).	Unpaired two-sided *t*-test	2 mg/kg	18 Gy	Tumor-cell repair mechanism is reduced	[125]
Lu–Au- NLS-RGD-anti-VEGF aptamer	Athymic male mice	6–7 weeks	MG	U87MG	Tumor size progression was significantly slower (*p* < 0.05)	Tumor size progression was significantly slower (*p* < 0.05)	Tumor size progression was significantly slower (*p* < 0.05)	ANOVA	75.19	80 Gy	Tumor development is inhibited by halting the formation of new blood vessels	[126]
Doc-NPs	BALB/c mice	4–5 weeks	GC	BGC823	Not significant in tumor doubling time	Not significant in tumor doubling time	Significant effect on tumor doubling time	*t*-test	5 mg/kg	15 Gy	Doc-NPs cause cell cycle arrest in G2-M phase whereas irradiation leads to ROS generation which induce DNA damage	[127]
AuNPs	C3H/HeJ mice	8–10 weeks	SCC	SCCVII	Time for doubling of tumor volume and survival time increased significantly (*p* < 0.04)	Time for doubling of tumor volume and survival time increased by 23 days and 37%, respectively	Time for doubling of tumor volume and survival time increased most significantly (*p* < 0.04)	2-sided Z-statistics test	1.9 g/kg	42 Gy	AuNPs boost the radiation treatment in a radioresistant squamous cell carcinoma	[128]
AuNPs	Syngeneic black B6C3f1 mice	NR	MBT	Tu-2449	No abatement of tumor growth	18% long-term survival	56% long-term survival	Log-rank (Mantel–Cox) survival analysis test with 95% CIs using GraphPad Prism^®^ software	4 g/kg	35 Gy	AuNPs combined with radiation cause about 53% (>1 year) tumor-free survival	[129]
NP@PVP + B + Pt	BALB/c mice	5–6 weeks	BC	EMT-6	Tumor growth was inhibited by 45.4%	Tumor growth was inhibited by 51.9%	Tumor growth was inhibited by 91.2%	Student’s *t*-test	4 mg/kg	5 Gy	The bismuth in NP@PVP + B + Pt functions as a radiosensitizer and increases the production of ROS, which damage DNA under X-ray irradiation	[92]

Abbreviations: NP = Nanoparticle; Dbait@H1 NPs = NP-DNA double-strand breaks bait (consisting of polycation polyethylenimine, cross-linked with β-cyclodextrin and conjugated with folic acid); Cat@PLGA_R837 = NP@Poly(lactic-co-glycolic) acid (PLGA) + water-soluble catalase (Cat) + hydrophobic imiquimod (R837), IPI549@HMP = PI3-kinase γ (PI3kγ) inhibitor (IPI549) + PEGylated HMnO_2_ (HMP); AuNPs = Gold-NPs; NPs-lncAFAP1-AS1 siRNA = Long noncoding actin filament-associated protein 1 antisense RNA1 with Small interfering RNA; Au@Tat-R-EK NPs = Gold + nuclear targeting peptide sequence (GRKKRRQRRRPQ) + peptide sequence (GFLG) + zwitterionic peptide sequence consisting of alternative glutamic Acid (E) and lysine (K); AuNPs@CRZ = AuNPs + Crizotinib; Lu–Au-NLS-RGD-anti-VEGF aptamer = Lutetium^177^ + Gold + Nuclear Localization Sequence-Arg-Gly-Asp + anti-VEGF aptamer; Doc-NPs = Docetaxel-NPs. NP@PVP + B + Pt = Polyvinylpyrrolidone with bismuth and cisplatin Gy = Gray (unit of ionizing radiation dose in the International System of Units); NR = Not reported; NSCL = Non-small cell lung cancer; PCa = Prostate cancer; CRC = Colorectal cancer; MC = Mammary carcinoma; BC = Breast cancer; LC = Liver cancer; ACC = Adenoid cystic carcinoma; MG = Malignant glioma; GC = Gastric cancer; SCC = Squamous cell carcinoma; MBT = Malignant brain rumor.

## 5. Uptake and Excretion of NPs during RT

The success of the NP-based RT largely depends on internalization of the nanostructured radiosensitizers into the tumor, along with clearance from the body to avoid any toxicity. Yi et al. explored the possible effect of X-ray irradiation on the uptake and efflux of NPs by cancer cells. It was reported that exposure to X-rays (6 Gy) enhanced the uptake of different nano-structured compounds, including melanin-coated CuS NPs with PEG, AuNPs, silica NPs, and HAS NPs, by 32%, 25%, 33%, and 20%, respectively, by 4T1 cells compared to without X-ray radiation. Furthermore, looking into the probable mechanism, irradiation causes a reduction in the G0/G1 phase while increasing the G2/M phase [130]. Cells show varying levels of endocytosis at different phases of the cell cycle: G2/M > S > G0/G1 [131,132]. Thus, increased G2/M phase after X-ray radiation increased the uptake of NPs by cancers cells [130]. Similarly, Davies et al. prepared nanosized liposomal doxorubicin and reported that RT increased the uptake of doxorubicin by two- to four-fold in the tumor [133]. However, traditional radiosensitizers such as cisplatin exhibited significant tumor uptake and excellent RT sensitization. However, the poor renal clearance caused kidney toxicity [134]. Larger NPs sized 20–100 nm accumulated in the spleen and liver where they lasted for few months, thus mediating long-term toxicity [135,136]. NPs smaller than 5 nm are eliminated shortly in vivo via renal clearance [137]. Zhang et al. fabricated polymer micelles with ultra-small AuNPs (PMG NPs) sized 1.9 nm. PMG NPs improved RT and were removed from the body without causing any toxicity [138].

## 6. Radiation-Induced Bystander Effects

The biological effects of radiation were believed to be restricted only within targeted tumor areas but have been found to affect adjacent non-targeted tissue surrounding the targeted area [139]. This event is called the radiation-induced bystander effect (RIBE), in which nearby non-irradiated cells act like radiation-exposed cells (Figure 3a) [140,141,142,143]. RIBE arises through signal transmission from irradiated to nearby non-irradiated healthy cells, direct cellular contact, or secreting of soluble factors in the neighboring area [142,144]. These bystander signals may alter genetic expression, exchange sister chromatid, cause genomic stability, damage DNA, reduce cell proliferation, and alter the translation process of non-exposed cells [142,145]. Major bystander signaling molecules are ROS, micro-ribonucleic acid (miRNA), and extracellular oxidized DNA [146,147], which reach surrounding healthy cells via binding to receptors or passive diffusion [147]. For instance, cells died by RT-released cell free chromatin particle (cfchp), which integrates into the bystander cells’ genome and causes DNA damage along with inflammation [148]. However, the abovementioned RIBE can be abrogated by cfchp inactivating agents, such as anti-histone antibody complexed NPs (CNPs). Kirolikar et al. reported that γ-rays (50 Gy) activated RIBE biomarkers, including H2AX, NFκB, IL-6, and caspase-3, in brain cells. Concurrent treatment with CNPs abrogated such RIBE biomarker activation and thus prevented cfchp-induced cell death (Figure 3b) [149]. Likewise, Zainudin et al. irradiated MCF-7 and hFOB 1.19 cells with a photon beam of 6 MV along with bismuth oxide NPs (Bi_2_O_3_ NPs). Exposure to ionization radiation plus Bi_2_O_3_ NPs did not induce ROS generation or DNA fragmentation due to RIBE. Therefore, the survival rate increased by 3–8% to both bystander cells [150]. Furthermore, Rostami et al. investigated the consequences of RIBE on MCF7 and QUDB cells after treating both cell lines with glucose-covered gold NPs (G-AuNPs) and 2 Gy of 100-kVp X-ray radiation. The results showed 13.2% reduced cell viability and an 11.5% decreased survival fraction in QUDB bystander cells compared to irradiated bystander cells without G-AuNPs. However, MCF7 cells did not show any RIBE upon radiation exposure [144].

## 7. The Effects of NPs-Based RT in Cancer Stem Cells

Cancer stem cells (CSCs) are common in most cancers; they cause metastases and function as a cancer cell reservoir, causing tumor relapse after CT, radiotherapy, or surgery [151]. CSCs’ ability to proliferate unlimitedly and their resistance to drugs pose great threats in cancer management [152]. CSCs are characteristically resistant to CT due to their quiescence, capacity to repair DNA, and ABC-transporter expression [153]. The self-renewal property allows them to extend tumor cell numbers after chemo- or radiotherapy [154,155]. Conventional treatments are not able to destroy CSCs; therefore, novel treatment strategies are highly demanded.

Fiorillo et al. developed graphene oxide (GO)-based nanostructures that were able to prevent tumor-sphere formation in six different cancer cell lines, including MCF7 for breast cancer, SKOV3 for ovarian cancer, PC3 for prostate cancer, MIA-PaCa-2 for pancreatic cancer, A549 for lung cancer, and U87-MG for brain cancer. They applied tumor sphere assay to measure the formation of tumor spheres to evaluate the effect on GO. The results suggested that GO targets the phenotypic prosperity of CSCs and reduces bona fide CSC numbers by inducing their differentiation and inhibiting their proliferation. More specifically, GO-based treatment inhibited several major signal pathways, including WNT- and Notch-driven signaling, STAT1/3 signaling, and the NRF2-dependent anti-oxidant response, together with inducing the differentiation of CSC, thus decreasing the general stemness [156]. Likewise, Yao et al. designed gastric CSCs targeting carbon nanotubes based on chitosan and loaded with salinomycin with hyaluronic acid (SWCNTs), which selectively eradicate gastric CSCs [157]. Later, Al Faraj et al. modified SWCNTs with CD44 antibodies and showed enhanced targeting of breast CSCs and promise in clinical studies [158].

## 8. NPs-Based RT Improving Phototherapy

NP-mediated phototherapy, including photothermal therapy (PTT) and photodynamic therapy (PDT), has exhibited promising efficiency in treating superficial and internal tumors [159]. However, the tremendous advantages of phototherapy come with some limitations. PTT has high efficiency in cancer treatment due to its controllable, accurate, and non-invasive properties [160,161,162]. PTT transforms photon energy to hyperthermia through photothermal agents (PTAs) [163]. However, a characteristic drawback of laser attenuation, PTA’s nonuniform distribution, and unwanted phototoxicity to healthy tissue limit PTT [164,165]. On the other hand, PDT employs photosensitizers to generate ROS in response to near infrared irradiation (NIR) [166,167]. Nevertheless, PDT also has limitation since a hypoxic TME hampers its application [168]. However, NP-based RT could be considered a promising adjuvant treatment to complement phototherapy [169]. Low energy and long wavelengths of near infrared light in phototherapy show limited penetration depth, while the high energy and short wavelength of X-rays or γ-rays are free from depth restrictions [170]. Furthermore, their combination reduces the radiation dose while increasing therapeutic efficiency [171].

Treating KB cells with 20 μg/mL of iron oxide coated with gold NPs (Au@Fe_2_O_3_ NPs) for 4 h, followed by PTT with irradiation at 808 nm, 6 W/cm^2^ for 10 min, and 6-MV X-ray irradiation (2 Gy), decreased cell viability at ~40% and ~20% compared to treating them with only RT and only PTT, respectively [172]. Further investigation revealed that such combination treatment in KB cells enhanced the expression of Bax/Bcl2 genes by five- and 6.67-fold, as well as HSP-70 protein by 1.84- and 2-fold, compared to radiation and laser irradiation, respectively [173]. Bax/Bcl2 genes are involved in regulating apoptosis and their overexpression causing cellular apoptosis [174]. In addition, HSP-70 gene overexpression denotes heating, oxidative stress, and inflammation, leading to cellular death [175]. Movahedi et al. reported similar results in KB cells with gold nanorods with folic acid (AuNRs-FA). AuNRs-FA (15 µg/mL), laser irradiation at 808 nm, 2 W/cm^2^, along with 6-MV X-ray irradiation, reduced cell viability to ~70% compared to single RT or PTT treatment [176]. Gonza’lez-Ruı’z et al. developed lutetium-177 labeled gold NPs with the nuclear localization sequence-Arg-Gly-Asp and an anti-VEGF aptamer nano system that significantly reduced U87MG tumor progression compared to only RT and only PTT under laser irradiation (532 nm, 1.19 W/cm^2^, 3.5 min) and 89 Gy of X-ray radiation [126]. Another in vivo study showed that PEG-[64Cu]CuS NPs inhibited the growth of anaplastic thyroid carcinoma and provided eight days more survival time than radiation or PTT treatment alone [177]. Liu et al. designed liquid-metal NPs consisting of metronidazole (MN) and GRD peptides (containing glycine-arginine-aspartic acid) linked to polyethylene glycol and polyacrylic acid (GRD-PEG-PAAMN@LM). Administering RGD-PEG-PAAMN@LM within HepG2 tumor-containing mice under NIR irradiation at a wavelength of 808 nm, 2.0 W cm^−2^ for 5 min and exposing them to 6 Gy for 2 min after 48 h almost abolished tumors after 14 days. Hence, the synergistic effect proved to be better than a single treatment. Regarding deep mechanisms, the authors explained that RGD-PEG-PAAMN@LM targeted ανβ3 integrin over expressive blood vessel walls of tumors and accumulated through endocytosis. RGD-PEG-PAAMN@LM produced excessive thermal energy along with ROS by means of PTT and PDT, which induced tumor apoptosis. At the same time, the MN part entered nucleus to enhance the X-ray radiation-mediated DNA damage [178]. Some other examples of combinational approaches are also described in the literature [179,180].

However, the highest synergistic effect of RT and phototherapy can be obtained when the two modalities have an optimal interval. Safari et al. synthesized NPs with gold and iron oxide cores and coated by alginate (Fe_3_O_4_@Au/Alg NPs). Exposure of KB cells to Fe_3_O_4_@Au/Alg NPs under 1 W/cm^2^ of laser irradiation for 5 min arrested the G2/M phase of the cell cycle and 24-h post-treatment with 6 Gy X-ray radiation showed maximum radiosensitivity with 68% apoptosis [181].

## 9. NPs-Based RT to Overcome Radioresistance and Drug Resistance

Radioresistance (RR) is a major constraint of RT, which leads to the recurrence of cancer [182]. Several improvements in radiotherapeutic approaches have been made out to increase the safety and efficacy while minimizing the RR of tumors. Among different approaches, combinational treatment of radiation with NPs stood as a prominent avenue to overcome RR [16]. Several mechanisms, including overexpression of DNA repair enzyme and anti-apoptotic proteins, contribute to the development of RR [183]. Moreover, progression of a hypoxic TME after irradiation is the one of the primary factors in RR development [184]. However, RR can be reduced by downregulating selective genes such as vascular endothelial growth factor (VEGF) expression that reduce hypoxia and result in a better radiotherapeutic response [185]. Li et al. constructed AuNPs encapsulating recombinant human endostatin (Au-RHES). RHES is a vascular angiogenesis–disrupting agent that normalized transient vascularization in the H22 xenograft model and diminished hypoxia [186]. Additionally, AuNPs tend to inhibit heparin-binding growth factors (HB-GFs) and basic fibroblast growth factor (bFGF), which are involved in tumor metastasis [187]. AuNPs encapsulating siRNA facilitated myelocytomatosis oncogene (c-myc) knockdown in cancerous HeLa cells [188]. In another study, Lee et al. discovered that high expression of low density lipoprotein receptor-related protein-1 (LRP-1) contributed to radio-resistant CRC [189]. LRP-1 plays a vital role in maintaining the TME and regulating cancer invasion due to its association with intracellular signaling and endocytosis of different types of cancer [190,191]. LRP-1 could be considered a marker protein for CRC’s RR [189]. These authors engineered a nano formulation consisting of human serum albumin possessing the LRP-1-binding peptide and B5 for tumor targeting, along with 5-FU and Cy7 (Cy7–B5–HSA–5-FU). Irradiation of Cy7–B5–HSA–5-FU resulted in the reversal of the radio resistance of CRC greatly and inhibited tumor growth significantly with treatment of 2 Gy for five days [189]. Furthermore, Li et al. reported that 1.7-nm platinum NPs increased γ ray radiation effects by >40%, thus overcoming RR in *Deinococcus radiodurans* [192].

Moreover, drug resistance is responsible for chemotherapeutic failure in 90% of patients with metastatic tumors and is still a significant obstacle to achieving success in CT [193]. Among a number of causes, hypoxia is one of the main reasons for drug resistance in tumor cells [194,195]. Therefore, releasing tumor cells from hypoxic conditions is essential to overcoming drug resistance. Nitric oxide (NO) has hypoxia-relieving properties. It can reverse cancer cells’ drug resistance and enhance their sensitivity to radiotherapy [196,197,198]. However, NO has a short half-life, and its sensitivity toward biological substances limits its clinical application [199]. On the basis of these findings, Zhang et al. fabricated a nanotheranostic agent by functionalizing bismuth with S-nitrosothiol (Bi-SNO NPs). Upon exposure to radiation, X-ray broke down the S-N bond and stimulated release of a great amount of NO. Bi-SNO NPs (300 µg mL^−1^) 36 nm in size under 5 Gy of X-ray irradiation on HepG2 cells released NO, subsequently damaged DNA robustly, and overcame drug resistance [200].

## 10. Clinical Trials

Clinical trials help to ensure the safety and effectiveness of newly formulated treatment modalities, which is why clinical trials hold immense importance in developing treatment approaches and assessing recommended drug doses (RDs) [201]. A phase I study was performed to determine the RDs and safety profiles of hafnium oxide (HfO2) nanoparticles, called NBTXR3, and external beam RT (EBRT) in adult patients with locally advanced soft tissue sarcoma (STS) (trial registration number: NCT01433068). NBTXR3 at a dose of 53.3 g/L was initially injected with 50-Gy EBRT for five weeks, and the dose was escalated. The result demonstrated that NBTXR3 at 10% of tumor volume was the recommended rose, and intratumoral injection resulted in a 40% median tumor shrinkage rate and a 26% median percentage of residual viable tumor cells [202]. Later, a multicenter, randomized, phase II/III trial in 180 STS patients was conducted to determine the safety and efficacy of radiation-assisted NBTXR3 and compared it with radiotherapy alone (trial registration number: NCT02379845). Patients were divided into two groups randomly: one group received EBRT alone (5 × 2 Gy by week) over 5 weeks, while the other group received NBTXR3-mediated EBRT at the same radiation dose and time. The NBTXR3-mediated EBRT group provided a two-fold greater pathologic response rate compared to EBRT alone. The results of the trial also showed that 39% patients in the NBTXR3-mediated EBRT group experienced emergent side effects, whereas 30% patients in the RT group experienced serious adverse events, such as injection site pain, postoperative wound complications, radiation skin injuries, and hypotension [203]. Other clinical trials data demonstrated similar results. However, numbers of registered clinical trials have not showed desired results and came up with some complications. However, overall, the NP-based RT was more effective than treatment with radiation only. Trial numbers NCT01946867 and NCT02901483 (head and neck cancer), NCT02721056 (liver cancer), NCT02805894 (prostate cancer), and NCT02465593 (rectal cancer) have been registered at clinicaltrials.gov but terminated afterward. However, overall, NP-based RT showed better efficacy than treatment with radiation only [204].

## 11. Limitations and Future Perspective of NPs-Based RT

Despite of the tremendous advantages of NP-based RT, there are still challenges that need additional development [205]. One of the primary challenges to improving the therapeutic efficiency of NP-based RT is the delivery of optimal concentrations of required NPs to cancer cells with minimal or no side effects [206,207]. Administration of NP-based radiosensitizers through the systemic route is deemed to have negative effects on other organs, in addition to tumors. This outcome reduces the patient’s general health, specifically when localized radiosensitization is required [207,208]. Therefore, the therapeutic approach to combining NP-based radiosensitizers with RT should be carefully determined due to radiosensitizers’ toxicity to normal tissues [208]. Another shortcoming of systemic delivery is that NPs that have extended circulating times may not reach the tumor during RT. Such a delay in treatment may lead to dose enhancement compared to localized delivery [209]. Delivering NP-based radiosensitizers via implantation has been used largely in RT. Nevertheless, this strategy is limited to only specialized cancers, such as those of the lungs, breast, prostate, etc., where spacers along with fiducial markers are commonly used [209,210]. Similarly, the inhalation route is only effective in lung cancer [209]. Therefore, further extensive research is essential to develop the present strategies for systemic delivery of targeted NPs.

However, the application of NP-based RT has great prospects, notably for implanting radiosensitizers for in situ dose painting in RT [211,212]. Brachytherapy Application with In-situ Dose-painting through Gold Nanoparticle Eluters (BANDAGE) is considered as an excellent treatment window to enhance therapeutic efficiency during brachytherapy [211]. BANDAGE is expected to elevate the survival rate of prostate cancer patients who need salvage therapy but have reached the dose limit of radiotherapy. Moreover, BANDAGE is also anticipated to be used in prostate-seed brachytherapy treatment to stimulate local radiotherapeutic effects without toxicity [205]. NP-based RT plays a vital role in the advancement of cancer treatment, including concomitant chemoradiotherapy. The ultimate aim of NP-based RT is to obtain greater efficacy with minimum side effects and reduced tumor reoccurrence, helping to improve cancer prognosis and prolong patients’ lives [213]. The ideal cancer treatment modality would require the perfect radiosensitizer, which has not yet been reported [214]. Hence, thorough research is required in this field. However, improvising a delivery route may help to achieve substantial dose enhancement with negligible amounts of radiation and drug toxicity.

## 12. Conclusions

RT is one of the main and most important treatment strategies against cancer with up to 50% of all cancer patients receiving radiation treatment. RT kills tumors selectively by delivering intense ionizing radiation. Such targeted treatment has been used to cure solid, as well as metastatic, tumors. However, RT has disadvantages, such as causing injury to nearby normal cells. Additionally, tumors distant from the radiation site receive less intense ionizing radiation. Furthermore, RR is one of main causes of the failure of RT and subsequent relapses of tumors. Numbers of cases of radiotoxicity have been reported in the literature. The introduction of NPs in RT provides a state-of-the-art strategy in the field of radiation treatment by not only bypassing the limitations and side effects of RT but also showing significant tumoricidal activity compared to radiation only or NP-based treatment against all types of cancer. NPs with high Z elements show robust radiation absorption cross-sections and thus are used as radiosensitizers in external RT. Apart from high Z materials, nanostructured radiosensitizing agents of different types are also applied in NP-based radiotherapeutic treatment. The incident photon interacts with the NPs’ atoms and stimulate the ejection of photoelectrons or auger electrons, which cause the destruction of cancer cells. The synergistic effects of NPs and ionizing radiation interfere with DNA-repair processes, also producing ROS, which damage tumor DNA. Moreover, their combination arrest the cell cycle of cancer cells and regulate the TME to inhibit tumor progression. Additionally, conjugation of NPs with radiation treatment restricts bystander effects and keeps surrounding non-cancerous cells undamaged. Such associated treatment reduces hypoxia, maintains the TME, and obstructs the tumor metastasis process, contributing to increased RR of tumors. Moreover, introducing nano RT with phototherapy diminishes phototoxicity, at the same time offering complementary treatment against cancer. Overall, NPs can be employed as radiosensitizers in RT, offering new opportunities to progress and improve radiotherapeutic treatment. Their synergistic effects develop the clinical efficiency of RT against various types of cancers. Therefore, more clinical research needs to be conducted to heighten the NP-based radiotherapeutic approach to cancer treatment.

## Figures and Tables

**Figure 1 cancers-15-01892-f001:**
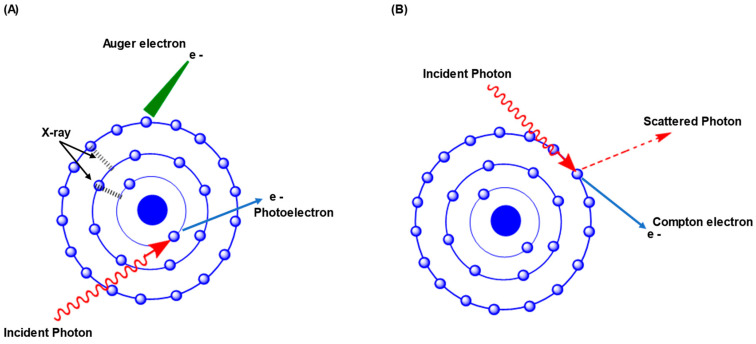
NP interaction with ionizing radiation. (**A**) The collision of incident photons with inner orbital electrons causes a photoelectric effect. The inner electron absorbs photon energy and ejects it as a photoelectron. Due to the ejection of photoelectrons, electrons from outer shell fill the gap resulting in the emitting of X-rays or the Auger electron. (**B**) On the other hand, the collision of incident photons with outer orbital electrons causes the Compton effect. The Compton electron absorbing photon energy is ejected from the atom, which may excite and ionize subsequent atoms. The photon loses a fraction of its energy and either continues on its course or is alternatively involved in further process, such as photoelectric effects or the Compton effect.

**Figure 2 cancers-15-01892-f002:**
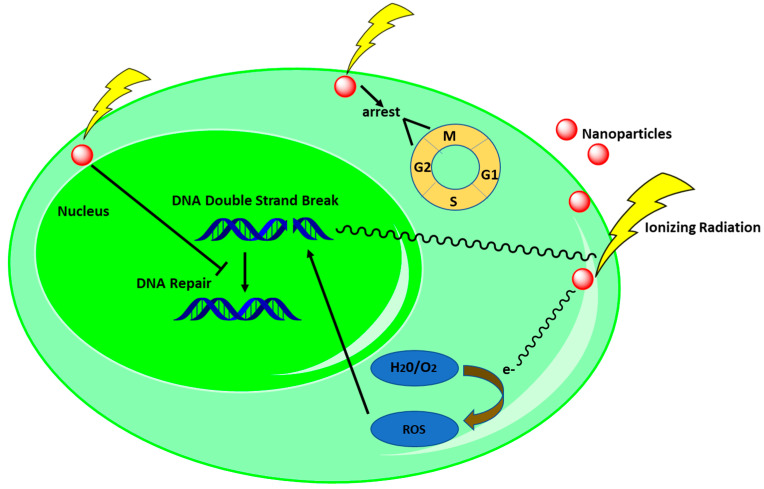
Biological responses of NPs aiding RT. Ionizing radiation causes direct damage to DNA, which is repaired by the cells’ own repair mechanism. However, NPs with RT interfere with the DNA-repair process, leading to cancer cell death. Moreover, ionization induces ROS generation, which damage DNA-killing cancer cells. In addition, radiation together with NPs arrests the G2/M phase of the cell cycle and results in cancer cells death eventually.

**Figure 3 cancers-15-01892-f003:**
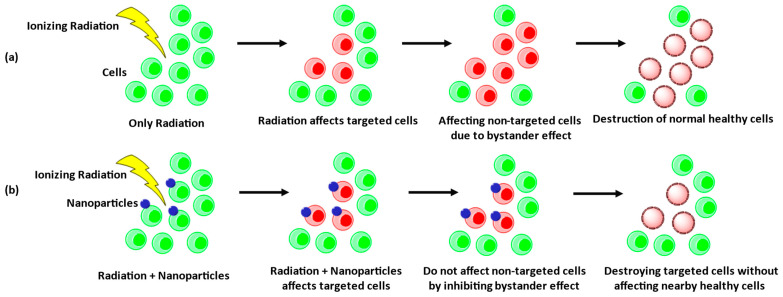
Radiation-induced bystander effects. (**a**) Irradiating cancer cells causes both targeted and non-targeted cell destructions. (**b**) Radiation along with NPs only kills tumor tissues, leaving the nearby healthy cells intact.

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
