# Peer review of "The Promise of Nanoparticles-Based Radiotherapy in Cancer Treatment"

_cancers, 2023, doi:10.3390/cancers15061892_

Round 1

Reviewer 1 Report

Haque et al. address the relevance of nano-particles based radiation therapy in cancer. However, this review has importance and timeliness.

Some suggestions may help for better impact.

1. A section of DNA repair response pathways and radiations may be included.

2. Implications of drug resistance and nano-particles based radiation therapy could be discussed.

3. Clinical trials on nano-particles based radiation therapy in cancer can be highlighted.

4. The effects of nano-particles based radiation therapy in cancer stem cells need to be highlighted.

5. ADME profile of nano-particles based radiation therapy should be critically discussed. 

Author Response

Dear Editor team,

Here is the response to the reviewer 1 corrections.

Thank you.

*****************************************

Reviewer 1:

Point: Haque et al. address the relevance of nano-particles based radiation therapy in cancer. However, this review has importance and timeliness. Some suggestions may help for better impact.

Response: We appreciate your optimistic feedback.

Point 1: A section of DNA repair response pathways and radiations may be included.

Response 1: Many thanks for your suggestions. In the revised manuscript, we have added DNA repair response pathway (in line 169-194). We hope it’s alright now. Thank you.

Point 2. Implications of drug resistance and nano-particles based radiation therapy could be discussed.

Response 2: Many thanks for your suggestions. We have included implication of nano-particles based radiation therapy against drug resistance (line 518-530) in the revised manuscript.

Point 3. Clinical trials on nano-particles based radiation therapy in cancer can be highlighted.

Response 3: Many thanks for your comments and suggestions. We have incorporated a new heading on clinical trials in the revised manuscript (in line 531-557). We hope it’s alright now. Thank you.

Point 4. The effects of nano-particles based radiation therapy in cancer stem cells need to be highlighted.

Response 4: Thank you for your comment. We have added a heading on the effects of nano-particles based radiation therapy in cancer stem cells (in line 415-438). Thank you.

Point 5. ADME profile of nano-particles based radiation therapy should be critically discussed. 

Response 5: Many thanks for your comment. We have discussed the asked point on under the heading of uptake and excretion of NPs during RT (In line 362-381) in the revised manuscript. Thank you.

Reviewer 2 Report

It was a manuscript about the application of nanoparticles as a radiosensitizer in the radiation therapy of cancer cells. Here are some comments on this study that should be considered before publication:

1.       The quality of the abstract is low. Please improve it.

2.       There are some grammatical mistakes in the text that should be corrected.

3.       Please rewrite lines 223-226.

4.       Figure 1 and Table 1 are not mentioned in the main text.

5.       Please add a subheading about the limitations and future perspective of the application of nanoparticles for cancer radiotherapy. Besides, please add another subheading about the combination use of radiotherapy and chemotherapy.

Author Response

Dear Editorial Team,

Here is the response to the reviewer 2 with the corrected manuscript.

Thank you.

************************************************

Reviewer 2:

Point: It was a manuscript about the application of nanoparticles as a radiosensitizer in the radiation therapy of cancer cells. Here are some comments on this study that should be considered before publication:

Response: We appreciate your optimistic feedback.

Point 1: The quality of the abstract is low. Please improve it.

Response 1: Many thanks for your suggestions. We have improved the quality of abstract in the revised manuscript. Thank you.

Point 2. There are some grammatical mistakes in the text that should be corrected.

Response 2: We apologize for the mistakes. We have corrected our grammatical mistakes throughout the revised manuscript. Thank you.

Point 3. Please rewrite lines 223-226.

Response 3: Many thanks for your comments and suggestions. We have rewritten the mentioned line (in line 282-288) in the revised manuscript. We hope it’s alright now. Thank you.

Point 4. Figure 1 and Table 1 are not mentioned in the main text.

Response 4: We apologize for this error. We have added figure and table number in the revised manuscript. Thank you.

Point 5. Please add a subheading about the limitations and future perspective of the application of nanoparticles for cancer radiotherapy. Besides, please add another subheading about the combination use of radiotherapy and chemotherapy.

Response 5: Many thanks for your comment. We have added a subheading about the limitations and future perspective of the application of nanoparticles for cancer radiotherapy in the revised manuscript (line 558-589). Also, we have added a subheading about the combination use of radiotherapy and chemotherapy (in line 85-108) in the revised manuscript. Thank you so much.

Round 2

Reviewer 2 Report

-